# CURATING PUBLICATIONS AS ARTEFACTS —
## EXPLORING MACHINE LEARNING RESEARCH IN AN INTERACTIVE VIRTUAL MUSEUM

**Beatrice Gobbo**
Department of Design
Politecnico di Milano
Milan, Italy
{beatrice.gobbo}@polimi.it

**Mennatallah El-Assady**
Department of Computer Science
University of Konstanz
Konstanz, Germany
{mennatallah.el-assady}@uni-konstanz.de

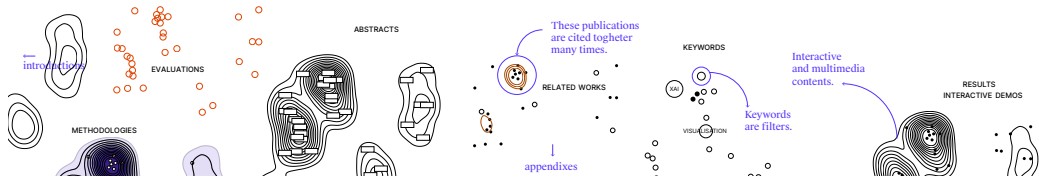

Figure 1: A sketch of the Machine Learning Interactive Virtual Museum. The *Exhibition Display*.

## ABSTRACT

The need for innovating scientific publications is felt across various research fields. In the last thirty years, the publishing process is accelerating, and new research is coming out every day. In Machine Learning, some attempts have been made to keep track of recent publications and to enhance articles for allowing authors to integrate multimedia contents. However, it is still hard to compare selected articles exhaustively. Thus, we envision a Machine Learning Papers Museum as a metaphorical solution for providing a digital space where users can explore publication collections in guided and serendipitous ways.

## 1 INTRODUCTION

The advent of digitisation, and the increasing trend to online-first presentations of publications, also due to virtual conferences, have made innovation in scientific publications increasingly necessary to undertake a digital transition that could improve the accessibility of scientific research (Lawson et al., 2016; Björk, 2005). Moreover, the regular release of new Machine Learning research is making the process of reading and reviewing papers increasingly tricky and complex. Because of their willingness to incorporate multimedia content, publications are more and more similar to complex artefacts; thus, considering the publication life cycle framework, we focused on rethinking how papers are published and disseminated, trying to outline the *discourse* that can arise from the interconnection between multiple contributions and authors. Thus, we envisioned an ideal setting for our proposal where Machine Learning papers are represented as *artefacts* in a virtual museum and connections between them are intensified through visual elements and layout (Figure 1). Readers are encouraged to explore the space either in serendipitous ways or following guided tours, customised according to the story they want to read. Firstly, in section 2 we describe the interdisciplinary nature of aiming at innovative scientific publications. Indeed, innovating scientific publications is a growing challenge in many academic contexts, not just in Machine Learning. Thus, we present a paragraph about the state of the art, in which we cite the most viable innovative models for scientific publications, focusing on the format in which papers are published and disseminated. In section 3, we describe conversations between actors during a publication process and how visual representation can translate fictitious dialogues. Then, we introduce the Machine Learning Papers Virtual Museum a metaphorical setting for our proposal. Afterwards, section 4 is dedicated to envisioning our proposal, which aims to dismantle a classic paper format and re-assemble it as if it were a piece of art. Finally, in section 5 we summarise our proposal and discuss future research challenges.

## 2 Innovating Publication Life Cycles: Disseminating & Exploring

Innovating scientific publication is not just a matter of suggesting new dissemination formats (Kim et al., 2018) but, overall, acting on a convoluted system that connects many actors, including researchers, publishers, editors, and research centres. Thus, it is necessary to consider that *publishing* is a process characterized by several stages that include both moments of personal production and occasions of *conversation* and interaction with the whole systems. Indeed, innovative scientific publications arise at various stages of the publishing process and across different research domains. (Lupo et al., 2021, forthcoming) A framework that allows us to identify the stages of the publication process across all areas of research is that of the *publication life cycle* proposed by Björk (2005)

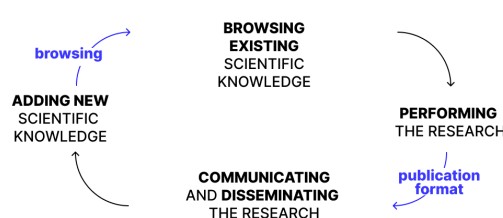

Figure 2: Visual summary of the publication life cycle from Björk (2005).

which outlines the "*communication value chain, from initial research to the assimilation of research results to everyday practice*." The main steps we will take into account include the browsing and analysis of the *existing scientific knowledge*, and the *communication and dissemination of the scientific knowledge* aimed to produce *new knowledge* expected to become integrated into the *existing source knowledge* (Figure 2).

### 2.1 Disseminating: Publication Selection, Aggregation, and Presentation

As mentioned before, efforts in innovating scientific journals touch various sides of the publishing process, such as the birth of *academic social networks*, the development of web-based scholarly journals, the adoption of new formats of articles including graphical abstracts, interactive PDFs or research data, the introduction of open distribution policies and the adoption of *Altmetrics* to complement traditional evaluation metrics (Kim et al., 2018).

In the mid-1990s, the advent of digitisation has led many journals to release both print and web-accessible versions. Indeed, the ability to read publications on-screen also allowed access via hyperlinks to related content (such as references), and this ensured a smoother and faster reading of content (Björk, 2011). However, the great innovation is found not so much in the transposition of content into digital format, but in the design of formats conceived primarily for digital publication. Distill (Figure 4 left) and Parametric Press are examples of web-based scholarly peer-reviewed journals born in the Machine Learning realm, where authors are encouraged to add lines of code, *explorables*, images and videos that allow for a more fluid and thorough reading of the content. While *Distill* focuses on Machine Learning research, *Parametric Press* addresses other domains and topics. A similar approach is devised by the Visualisation for AI Explainability Workshop where the formats to be submitted are interactive blog posts. In the Arts and Design fields, a comparable strategy is identifiable in the format promoted by the Dialectic Journal where editors welcome original visual essays and narratives containing photographs, illustrative work, typographic elements, tables, charts, diagrams maps and (soon) videos and animated clips. Moreover, in the medical and biological domains, efforts are already being made to move beyond the static PDF and accelerate the dissemination process employing publication formats dedicated to minor findings (Micropublications) and video-based experiments (Jove). In Digital Humanities, Stanford University Press (Figure 3 right) proposes an online journal dedicated to digital products able to "*confer the same level of academic credibility on digital projects as print books receive*." Here authors can submit digital projects in a website format, showcasing historical narratives, interactive maps and data visualisations (Mullaney, 2018; Tsing et al., 2020).

The examples above come from various fields that allow both to employ innovative formats and browse publications available in each journal. Indeed, each of the previously presented journals also has a part dedicated to exploring available publications.

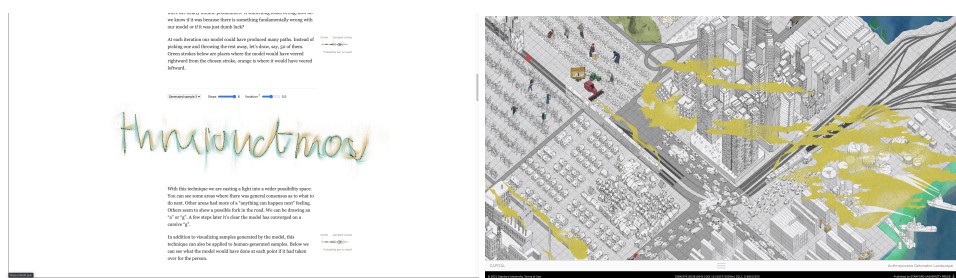

Figure 3: A sample of innovative publishing formats. **Left**: A Distill article about Handwriting with a Neural Network. **Right**: An example from Stanford University Digital Press.

## 2.2 EXPLORING: RE-ENTERING THE PUBLICATION CYCLE

Exploring existing works is a relevant action in the publication life cycle, as each new paper builds on the robustness of this process. Thus, innovating academic publications also means innovating tools that allow for browsing literature according to specific and bespoke parameters. On the one hand, the previously mentioned journals propose innovative formats that can support various types of content, but, on the other hand, users can browse and explore the published collections in a very traditional manner by scrolling lists of items grouped by topics, issues or editions as depicted on the left in Figure 4.

In the last decade, digital platforms for browsing scientific publications mushroomed, such as *academic social networks*, mobile applications, and repositories of online conferences. *Academic social networks* are both catalysts of bonds with other researchers and tools for exploring publications related to specific research area or topics, such as ResearchGate or Academia, where scholars can build their customised feed adding research interests, institutions or people. Moreover, Apple and Google Play stores showcase mobile applications meant to help researchers explore, find, and collect publications across various venues or journals, such as Researcher App. However, those examples do not boast editorial curatorship related to a single venue or journal but are universal tools that researchers in various disciplines can use.

Innovative strives on explorations of existing academic knowledge come from specific conferences or journals, as in IEEE VIS Conference, which at its first virtual conference (due to the COVID-19 pandemic) welcomed online participants with an interactive exploration of the proceedings (Figure 4). However, if, on the one hand, literature - for instance, in the data visualisation field - offers examples of dynamic publications browsers (Kucher & Kerren, 2015; Isenberg et al., 2017), on the other hand, it seems there are no journals that offer both an *innovative* browser for published articles and an *original* publication layout. Given the state of the art, designing new ways for *exploring* existing scientific knowledge and *devising* new publication formats are pivotal issues for rethinking scientific publication models in Machine Learning.

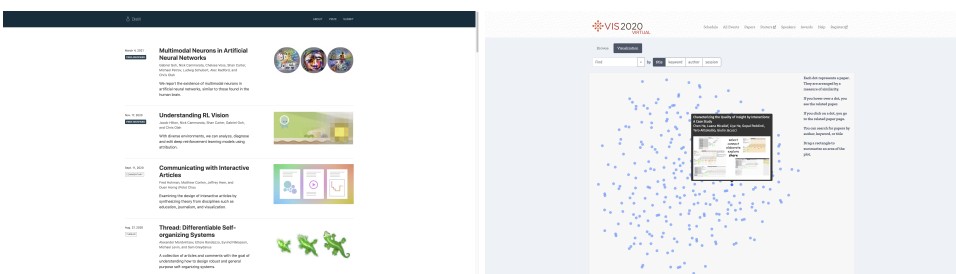

Figure 4: **Left**: A traditional way of presenting articles from Distill. **Right**: The IEEE VIS Virtual Conference example that proposes an interactive visualisation for browsing papers.

## 3 RESEARCH AS CONVERSATIONS AND ARTEFACTS IN A VIRTUAL MUSEUM

In the following paragraphs, we outline how research papers could be seen as artefacts and conversation triggers. On the one hand, a publication (its related works section, abstract, keywords, pictures, methodology and results) could be intended as a group of building blocks. On the other hand, a Machine Learning publication is composed of multimedia elements, pictures and visualisations that make it increasingly closer to an artefact.

### 3.1 PUBLICATIONS AS DISCOURSE CONTRIBUTIONS

The publication life cycle itself, including the literature browsing and its layout, is a *communication design process* devoted to disseminating the research content. Indeed, according to semiotic theory, effective communication requires understanding and cooperation from the reader (Patriotta, 2017). Thus, it implies adopting certain conventions in academic writing that allow scholars, students or practitioners to orientate while reading and make connections between various manuscripts. For this reason, publication formats use to be well-defined, split into *sections* whose content is specific and aimed at prompt understanding by the reader (Patriotta, 2017). Indeed, publication collections could be envisioned as conversations between actors (readers, authors and editors) that can be more or less broad, depending on the issues covered by each contribution.

In Machine Learning, tools visualising connections between publications aimed at outlining relationships among authors and works have been designed: Connected Papers provides tools for building customised networks connecting papers related to a chosen starting seed. Thus, users can build a literature review and explore new fields without missing any critical publication. Other cases outlining relations among publications could be found in *visual surveys* (Gobbo, 2021, forthcoming) focused on visual relations among teaser pages of a collection of selected papers. The examples above are witness of a trend toward designing *conversations* with an increasingly important visual component. Moreover, literature shows that some attempts to summarise publications' *sections* in visual form to make faster the reading process and facilitate comparison and critical readings have been done (Strobelt et al., 2010).

### 3.2 PUBLICATIONS AS ARTEFACTS

In Machine Learning, publications are more and more comparable to complex artefacts, composed of different media types that allow readers to read the publication at different levels of depth (focusing on pictures, on interactive elements or animations). Indeed, pictures, interactive elements, videos, math formulas, and animated illustrations with written text enable readers to understand key elements. For that reason, in Machine Learning, due to the complex mixture of media and interactions, it is necessary to design tools that allow comparative fruition of multimedia artefacts. However, in the last years, the growing relationship between computational methods and humanities has encouraged the design of interactive tools aimed at facilitating and augmenting the exploration of large corpora of art or literature pieces, going beyond the old-fashioned *search-centric and grid-based interface* and enlarging the range of interactions proposing principles such as serendipity and *generosity* in connection with information design (Windhager et al., 2019). An Ocean of Books (Gael, 2020) (top Figure 5) is an interactive poetic map connecting thousands of literature pieces, where each island represents a single author, and each city represents a book. The user is free to randomly explore the landscape, starting from a panoramic view from the top, where all the authors are clustered according to literary genres. Beyond Scroll and Screen (Sugrue, 2020a) (bottom-left Figure 5) and Magnify Miniature (Sugrue, 2020b) are examples of how a collection can be explored exploiting Computer Vision API technology. Each artefact correlates with a list of labels that can help users filter the collection and focus on recurring items or objects depicted. The T-SNE Map (Cyril Diagne, 2019) by Google Arts Experiments (bottom-right Figure 5) offers a 3D digital environment built on top of a T-SNE visualisation, which provides users with an interactive collection browsable both through guided or serendipitous tours. The examples gathered from different perspectives (Sections 2 and 3) suggest that browsing and assembling publication's building blocks is a viable solution for rethinking Machine Learning papers visually. On the one hand, some efforts have been made to provide interactive formats incorporating complex visual items, but browsing and exploring publications is still confined to traditional reading activity, where sections appear one after another. On the other hand, interactive surveys, automated literature review generators, and Digital Humanities

projects allow for building interactive spaces for exploring cultural collections.

Thus, since (Machine Learning) publications are *cultural collections* too, we envision the *Art Museum* as a viable metaphorical solution to rethink Machine Learning papers by taking inspiration from Arts and Design.

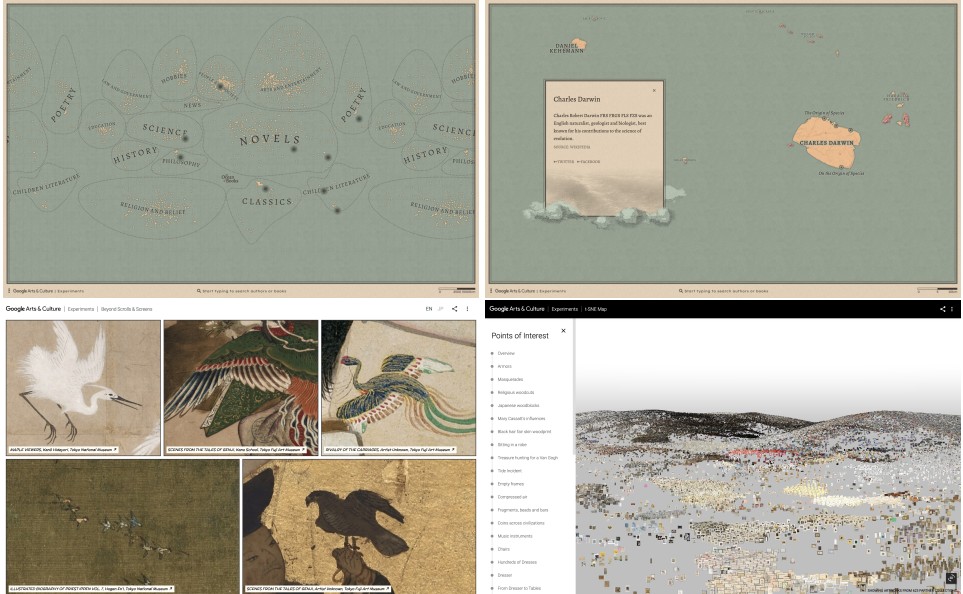

Figure 5: **Top**: The panoramic view showcasing clustered elements according to related literary genres and a focus on a selected author. **Bottom-left**, a screenshot from the Beyond Scroll and Screen project. **Bottom-right**: The T-SNE map view.

## 3.3 Publications in a Virtual Museum

Art Museums visitors, such as Digital Collection explorers, are welcomed to seek artefacts following guided tours or in serendipitous ways (Thudt et al., 2012). Guided tours usually provide critical readings, bespoken pathways of exploration, suggestions for making connections among art pieces available in different rooms. Serendipitous exploration transforms visitors into *flaneurs*: "*curious, creative, and critical information seekers*" (Dörk et al., 2011). Hence, in this paper, we propose a solution for innovating Machine Learning publications that embrace the above-described activities. As mentioned before (Section 3.2), due to their growing multimedia nature, Machine Learning publications can be compared to conventionally structured artefacts (Section 2). Thus, we foresee a way of devising, writing and exploring paper that improves the reading process and smoothly connects the dissemination and browsing steps (Figure 2) of the publication life cycle.

As artefacts in a digital collection (Figure 5), Machine Learning papers can be displayed in the space according to their mutual similarities and explored by comparing shared sections. As museum exhibits use to range over rooms, our publication gallery will range over sections. Thus, we envision a digital space composed of metaphorical rooms with *building blocks* depicting, for instance, all the related works sections. Rooms can have different shapes, according to the relationships identified between their contents: related works' room could be shown as a co-occurrences graph were works cited together are clustered. While the overall landscape is composed of as many rooms as building blocks identified in the papers' structures.

## 4   Curating, Dismantling, Enriching, Setting Up, and Navigating

For designing the Machine Learning Paper Museum, it is crucial to arrange a *vademecum* (or a handbook) for compiling new contributions; however, if authors provide organised information (Strobelt et al., 2010) (tags, labels, summaries, and interactive elements), some of the steps we mention later will become obsolete. Indeed, we foresee a malleable format that both encourages authors to correlate written sections with multimedia contents, and helps readers and researchers study and compare publications. Essentially, designing a *Machine Learning Papers Museum* requires curating, dismantling, enriching, setting up and navigating actions.

**Curating** means selecting and labelling publication collections. Once papers have been accepted for publishing, it is necessary to group them according to shared topics and labelling sections to lay the foundation for the *dismantling* process.

**Dismantling** means identifying papers' building blocks and essential elements for visually representing them in an exploratory space.

**Enriching**. To enhance the visiting experience, authors might be asked to integrate additional materials to enrich the publication.

**Setting Up** means locating items in the space, a crucial action aimed to visualize contents and create an exploratory environment.

**Navigating**.  As also happens in physical art museums, we wanted to foresee the opportunity to organize guided tours where it will be at the users' will to turn on or off a level of annotations that highlights a guided or critical reading of the space.

### 4.1   Curating: Selecting and Coding Collections

As happens in physical museums, curatorship is critical to the final display of the artefacts. Curating papers means selecting, labelling publication collections according to their topics, coding them, and envisioning potential relations. Since this step is preparatory to the next, it is necessary that editors and committee members check if publications already present a building blocks structure or if a further inspection is needed.

### 4.2   Dismantling: From Paper Building Blocks to Artefacts

Dismantling requires a format including various type of building blocks, namely the **rooms** of our *Machine Learning Papers Museum*. This process could be carried on computationally (extract figures, links, text, formulas, algorithms, summarise each building block), or authors should add these contents in a format directly. Usually, ML papers are composed of:

**Abstract/Teaser Image**: Essential elements to get an overview of the paper. The format would also request a *graphical abstract: a single, concise, pictorial and visual summary of the main findings of the article.*

**Related Works**: The room devoted to literature review building blocks.  The format we envision would require authors to provide a networked graph of relationships between authors and publications, based on a nodes-edges table.

**Methodology**: Our format would also accept methodology sections described in a diagrammatic form to promote visual elements beside texts.

**Evaluation**: In this room, besides small test descriptions, readers could also find recorded user test sessions and interviews.

**Results**: This room will contain final outputs, usually conceived as interactive tools or demos correlated with text, code snippets and math formulas.

**Further Works or Research Opportunities**: Further works and research opportunities are junctions with correlated future publications.  This room would contain keywords and unambiguous statements to be linked to forthcoming papers.

### 4.3 ENRICHING: CONNECTING OUTSIDE RESOURCES FOR REFINED STORYTELLING

For boosting the experience, authors will be asked to provide additional material such as video interviews about intentions, design processes, pitfalls, lessons learned and external links for accessing further information. This step is essential because additional material would allow readers to understand how to improve further research and not make the same mistakes. Moreover, connecting external resources will improve storytelling.

### 4.4 SETTING UP: LAYING OUT THE ARTEFACT DISPLAY

Setting up a collection of artefacts requires reflection on space constraints (which artefacts to display and which to hide in an archive?) and on the relationships among items. Indeed, over time and as new publications and issues are published, the space occupied by building blocks will grow. Thus, we envisioned two layouts: the *exhibition* and the *gallery*. Both of them are the embodiment of relations between authors and catalysts of conversations between authors and readers.

The **exhibition display** (Figure 6) provides an overall perspective, where each paper is *dismantled* and displayed in the corresponding room. The same type of blocks related to each contribution will be visualised according to similarity. Users will be free to explore the space *serendipitously* by panning, zooming and filtering according to keywords or topics. Moreover, each item will correlate with information referring to the paper it comes from.

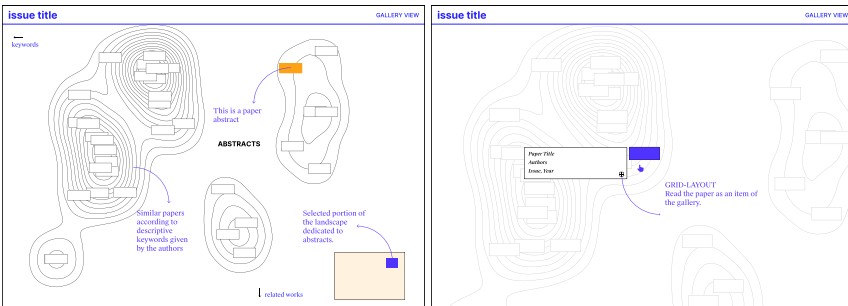

Figure 6: **Left**: Users can explore the *exhibition*. **Right**: A *tooltip* gives more information.

The **gallery display** depicted in Figure 7 has a grid layout that allows readers to scan a single paper in its ensemble of building blocks and compare the same building block throughout many publications. The grid arrangement allows readers to have many building blocks at their fingertips, ensures faster cross-reading, and fuels unexpected connections. In our idea, users will be free to compare multiple rooms or multiple papers, adapting grid size to their needs.

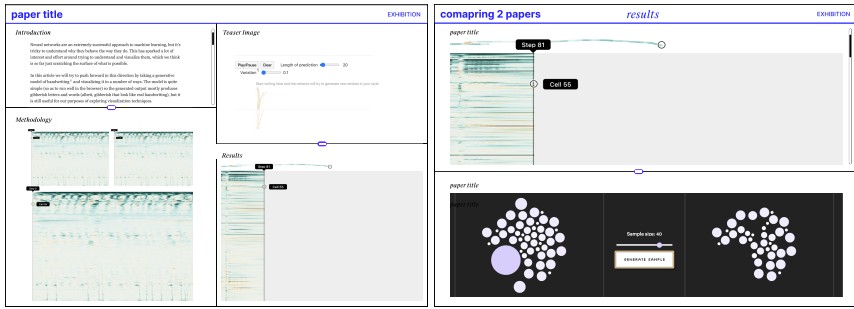

Figure 7: **Left**: An example of *gallery* view where users compare building blocks. **Right**: A comparison between two publications. These pictures come from Four Experiments in Handwriting with a Neural Network (Carter et al., 2016) and The Myth of The Impartial Machine (Feng et al., 2019)

## 4.5 NAVIGATING: TAILORED GUIDED TOURS THROUGH THE SPACE

In Digital Humanities, expert annotations are valuable tools for helping users orient in space. In a traditional museum environment, curators and reviewers provide critical readings and guided tours throughout exhibitions and galleries, letting users orient in the space. On our Virtual Museum for Machine Learning Papers, guided tours provide multiple information: firstly, they welcome readers into space, explaining how it is composed and how to orient themselves. At a later time, instead, they can provide additional information about the artefacts. For instance, they can explain to readers the reason why contributions are linked together, highlighting similarities and dissimilarities, they can support critical readings, connecting publications and making more explicit the *discourse* arising from connections and, besides, they can help to delineate state of the art from a selected cluster of publication. Guided tours can focus on comparing specific building blocks providing, for instance, a close reading on teaser images. Moreover, they can be tailored to users' need, providing visits according to their expertise and knowledge. Given these assumptions, we picture guided tours as layers of written or audio annotations that users can read or listen to (see section 5) at their first visit and as virtual thematic tours that users can access anytime during the virtual museum exploration (Figure 8). (Gobbo & El-Assady, 2021, forthcoming)

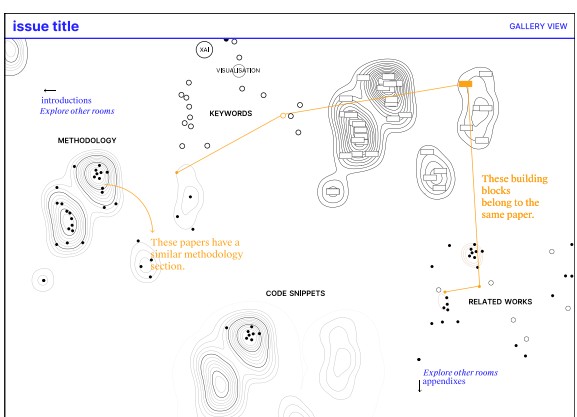

Figure 8: A sketch of editor's notes guiding the users through space. In blue a sample of notes helping users orient. In orange, an example of annotation that make connections among different rooms.

## 5 CONCLUSION AND FUTURE WORK

In this paper, we identified *browsing* and *dissemination* as crucial steps of the publication life cycle for renewing Machine Learning papers, presenting some remarkable case studies. Publications prompt authors-readers conversation and stimulate the use of multimedia contents, making them more and more similar to interconnected and complex artefacts. Thus, by taking inspiration from Digital Humanities and the visualisation of Cultural Heritage, we envisioned the *Machine Learning Paper Museum*: an evocative space that, considering both the conversational and multimedia nature of Machine Learning papers, encourages out of the box ways of browsing and disseminating scientific knowledge. The described idea is suitable for Machine Learning publications because, firstly, authors need to include more and more multimedia contents, and, subsequently, the regular release of new research is making the process of reading and comparison increasingly tricky. Thus, thanks also to filtering options granted by keywords, the *Machine Learning Paper Museum* is appropriate for accepting contributions from various Machine Learning sub-fields.

Moreover, dismantling traditional papers in modular building blocks allows us to put together and connect both short and long papers, showing a more comprehensive collection and arousing considerations about the State of The Art. We think that our Machine Learning Paper Museum would help the conception of new State of The Art Reports (STAR), providing a tool for comparing multiple contributions. In addition, *guided tours* could also be exploited for showing the state of the art of subsets of the collection, becoming themselves a STAR paper format.

## ACCESSIBILITY STATEMENT

The solution we envisioned in this paper would provide a significant step forward in terms of accessibility, emulating physical art museum that is working on welcoming users with vision or hearing disabilities and considering the Web Accessibility Content Guidelines.

First of all, proposing a multimedia approach ensures usability by a wider audience: the employment of video, audio or visual elements — correlated with alternative text and subtitles for screen reading — limits linguistic gaps. Besides, special attention will be paid to design colourblind-friendly visualisations and create a dyslexia-friendly virtual space, curating colours and typography.

Guided tours (and the digital infrastructure behind them) are also valuable solutions for adding audio-captions and reinforce the narrative nature of the curatorial aspect of our virtual museum.

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
