# OpenReview forum: "Curating Publications as Artefacts — Exploring Machine Learning Research in an Interactive Virtual Museum "
_ICLR.cc/2021/Workshop/Rethinking_ML_Papers — Rethinking ML Papers - ICLR 2021 workshop Oral_

### Official Review · AnonReviewer2 · 2021-03-28
**Interesting and well-researched, but presentation and specificity could be improved**

**Accessibility:**

Score of 3 (Neutral): Submission proposes methods to improve accessibility, but the level of intended accessibility is not well-articulated. Also, the limitations and exceptions are not stated.

**Litreview:**

Score of 4 (Strong): The submission directly differentiates itself from previous works and formats.

**Problemstatement:**

Score of 4 (Strong): The submission sets a very strong example of how to address the problem, which should be relevant to the workshop themes.

**Relevance:**

Score of 4 (Strong): The submission directly addresses a theme of the workshop, and does so in a very professional manner.

**Results:**

Score of 2 (Needs Improvement): Submission shows a poor level of clarity, novelty, coherency, and interactivity.

**Reviewerconfidence:**

4 - I do not think I am more or less qualified than a typical ML researcher to review this paper.

**Reviewtext:**

This paper proposes creating a "virtual museum" of machine learning papers that treats individual portions of papers as "artifacts." The proposed virtual museum would allow visitors to better make connections between components of different papers, and allow visitors to browse in either a "guided tour" format or through free exploration.

The paper connects to some previous literature about the scientific publication and communication process, and gives several examples of previous "alternative" publication formats for ML papers that would integrate well into the proposed virtual museum. It gives some level of concrete detail on how the proposed format would be constructed, for example with mock-ups of "rooms" of the virtual museum.

The biggest limitations of the paper, in this reviewer's view, is that the writing and organization of the paper has room for improvement, and the specific details of how the museum would be constructed could be better explicated. In particular:
* The introduction suggests that "guided tours" could be "customized according to the story [the viewer] want[s] to read." How should these guided tours be constructed, and what is the role of authors/reviewers/editors in constructing them? The first paragraphs of Sections 4 and 4.1 raised similar questions in my mind.
* The paper suggests integrating components of different papers in rooms corresponding to typical paper sections. This seems like it would require authors to rethink their papers so that each section could stand on its own, free of context from the rest of the paper. What are the drawbacks (or benefits?) of this?
* Section 4.2 suggests that related work sections be presented as a graph structure. How would these graphs be combined if the edges in these graphs represent different ways papers can be related? (e.g., competing techniques, building blocks, motivating works, etc.)

**Score:**

Accept: The reviewer believes the submission provides a novel and reliable scheme to improve science communication but needs improvement.

---

### Official Review · AnonReviewer1 · 2021-04-01
**ML Paper Museums**

**Accessibility:**

Score of 5 (Exceptional): Submission identifies and articulates accessibility matters, provides justifications for the proposed paradigm, and declares the limitations.

**Litreview:**

Score of 5 (Exceptional): The submission directly differentiates itself from previous works and formats, and provides enough structure (code, instructions, etc.) for the submission to itself be a new standard.

**Problemstatement:**

Score of 5 (Exceptional): The submission states a well-known problem relevant to the workshop, and sets what could be a new standard in the field when it comes to addressing it.

**Relevance:**

Score of 5 (Exceptional): Like (4) but does so with multiple themes of the workshop.

**Results:**

Score of 5 (Exceptional): Submission has an excellent design and all criteria are addressed. Conclusions, practical/theoretical implications are well articulated.

**Reviewerconfidence:**

4/5: Based on the paper's extensive interdisciplinary and well developed argument, I am confident in my review/decision to accept the paper to the workshop!

**Reviewtext:**

The authors propose a Machine Learning Papers Museum for exploring and disseminating ML research. The authors ground their proposed idea in the publication life cycle and focus on the need for discourse. A strength is the paper’s integration of interdisciplinary research on science communication, discovery, information curation, and existing examples of movement in this direction. A question I had based on the paper’s framing around “curation” was how authors think curation could affect the larger story told about ML research. Specifically, a great deal of art critique and history takes up how art pieces are juxtaposed to one another. Taking the museum curation idea a bit further, a curator’s job is to develop a compelling angle or narrative based on art in archive and new acquisitions. How do the authors envision this process arising? Whose narrative is portrayed and what are the potential benefits and dangers?
Overall, the paper makes a compelling argument and should be a great addition to the program!


**Score:**

Strong accept: The reviewer has a strong enthusiasm to apply the proposed framework in their work.

---

### Meta-Review · Area_Chair1 · 2021-04-01

**Recommendation:** Accept
**Confidence:** 4

**Metareview:**

The paper describes a proposal for presenting ML research in a way inspired by museum layouts and digital humanties methodology. The "visitors" to the "museum" would be able to take "guided tours" or to "browse" the environments constructed to connect the papers by keywords, by methodology, by "related papers" annotation.

The reviewers agree that the paper is highly relevant and the contribution is distinct, but they both have questions that should be addressed in the camera-ready version. In particular, the curation mechanism invites the following questions:

- who would do the curating? Clearly, the curators would have to be domain experts, and composing a quality exhibition would be tantamount to writing a survey of a subfield -- so this is too big of an effort to be undertaken as volunteers.
- how could we ensure fair participation and representation for the authors outside of the rich-get-richer loop?
- who would get to set the "narrative"? What if the authors disagree with the angle chosen by the curators?
- would the papers would have to be written and structured any differently to accomodate the "museum"?
- what kinds of connections would the related literature graph need to provide (shared methodology, same task, same field, same model, same data, same language)?

---

### Decision · Program_Chairs · 2021-04-01

Accept (Oral)